# Kagomerization of transition metal monolayers induced by two-dimensional hexagonal boron nitride

Hangyu Zhou [1,2,3,4] ✉, Manuel dos Santos Dias[1,5,6], Youguang Zhang[2], Weisheng Zhao [3] ✉ & Samir Lounis [1,5] ✉

The kagome lattice is an exciting solid state physics platform for the emergence of nontrivial quantum states driven by electronic correlations: topological effects, unconventional superconductivity, charge and spin density waves, and unusual magnetic states such as quantum spin liquids. While kagome lattices have been realized in complex multi-atomic bulk compounds, here we demonstrate from first-principles a process that we dub kagomerization, in which we fabricate a two-dimensional kagome lattice in monolayers of transition metals utilizing an hexagonal boron nitride (h-BN) overlayer. Surprisingly, h-BN induces a large rearrangement of the transition metal atoms supported on a fcc(111) heavy-metal surface. This reconstruction is found to be rather generic for this type of heterostructures and has a profound impact on the underlying magnetic properties, ultimately stabilizing various topological magnetic solitons such as skyrmions and bimerons. Our findings call for a reconsideration of h-BN as merely a passive capping layer, showing its potential for not only reconstructing the atomic structure of the underlying material, e.g. through the kagomerization of magnetic films, but also enabling electronic and magnetic phases that are highly sought for the next generation of device technologies.

The kagome lattice is a two-dimensional (2D) network of corner-sharing triangles which offers a fertile ground for exploring geometry, correlations, and topology in condensed matter physics[1]. The frustration arising from the constituent triangular units leads to the emergence of frustrated magnetic order[2–4] and spin liquid phases[5,6], as well as exotic electronic structures[7,8]. The frustrated geometry leads to the emergence of flat bands associated with electron localization, which are expected to increase the correlation effects and support many-body electronic phases and novel topological phases including ferromagnetism[9–11], unconventional superconductivity[12,13], and fractional quantum Hall states[14,15], to name only a few. Magnetic kagome lattices, especially in the 3d transition metal compounds, display intrinsic anomalous Hall effects driven by various mechanisms[16–19].

In recent years, 3d transition metal kagome structures have been reported for several materials[20], including $CoSn$[21,22], $FeSn$[8,23], $Fe_3Sn_2$[3,24,25], $Mn_3Sn$[19,26], $Co_3Sn_2S_2$[4,27,28], and $AV_3Sb_5$ ($A$ = K, Rb, and Cs)[13,29–33]. While these systems are interpreted as demonstrations of the physics of 2D kagome lattices, their bulk nature makes the connection less transparent[34]. Moreover, monolayer vanadium-based kagome metals are predicted to be thermodynamically stable and to have distinct properties compared to their three-dimensional bulk forms[35], demonstrating the importance of dimensionality for kagome

[1]Peter Grünberg Institut and Institute for Advanced Simulations, Forschungszentrum Jülich & JARA, 52425 Jülich, Germany. [2]School of Electronic and Information Engineering, Beihang University, Beijing 100191, China. [3]Fert Beijing Institute, School of Integrated Circuit Science and Engineering, Beihang University, Beijing 100191, China. [4]Shenyuan Honors College, Beihang University, Beijing 100191, China. [5]Faculty of Physics, University of Duisburg-Essen and CENIDE, 47053 Duisburg, Germany. [6]Scientific Computing Department, STFC Daresbury Laboratory, Warrington WA4 4AD, United Kingdom. ✉e-mail: h.zhou@fz-juelich.de; weisheng.zhao@buaa.edu.cn; s.lounis@fz-juelich.de

materials. Therefore, materials that indeed comprise just a single kagome layer can serve as an ideal platform for the prototypical realization of the electronic structures and novel phenomena arising from reduced dimensionality. However, to achieve this goal we must overcome the scarcity of genuinely 2D kagome materials.

Even though we are interested in achieving a truly 2D kagome material, in reality it will have to be deposited on a surface and likely covered with some other protective material. In this context, two popular and quite distinct encapsulation materials are graphene and *h*-BN. While graphene is a semimetal, *h*-BN is a large band gap insulator which makes it a suitable substrate and encapsulation layer rather than a functional material for spintronic applications[36–40]. However, recent studies have shown that *h*-BN can have a substantial impact on the encapsulated material, leading for instance to sizeable Rashba effect and Dzyaloshinskii-Moriya interaction (DMI) when interfaced with magnetic materials[41–43].

In this work, we show that it is feasible to realize a monolayer kagome lattice with the assistance of *h*-BN, a novel structural rearrangement which we term kagomerization. Using density functional theory (DFT), we demonstrate that *h*-BN can serve a functional role in facilitating remarkable structural reconstructions of transition metals. In particular, we find that kagomerization is rather universal when the epitaxial 3*d* transition metal monolayer (abbreviated as the TM layer) is grown on heavy metal (HM) fcc (111) surfaces, as unveiled for Pt, Au, and Ag substrates, as illustrated in Fig. 1a. We rationalize this general behavior by analyzing the contributions from *h*-BN and substrates. Subsequently, we focus on the structures containing ferromagnetic (FM) elements Fe, Co, or Ni. We identify electronic hallmarks expected for the kagome lattice such as Dirac cones and flat bands. Moreover,

we investigate the effect of this kagomerization on the magnetic properties and magnetic interactions. The frustrated Heisenberg exchange interactions, competing with DMI and magnetocrystalline anisotropy, lead to the formation of topological spin textures, such as skyrmions and bimerons, which are promising for the next generation of data storage devices[44–46]. The spin textures discovered in kagome monolayers, in particular the bimerons, manifest a plethora of magnetic phases responsive to applied magnetic fields. These discoveries offer new opportunities for further experimental exploration aimed at realizing an individual kagome lattice and stabilizing complex spin-textures.

## Results

### Kagomerization induced by *h*-BN

The considered structure consists of a 3*d* TM monolayer grown on fcc(111) HM (Pt, Au, Ag) substrates with a capping *h*-BN layer, as depicted in Fig. 1a. The experimental lattice constant for *h*-BN is 2.504Å[47], which is substantially smaller than the experimental lattice constant of the surface fcc(111) unit cell of Pt (2.775Å[48]), Au (2.884Å[49]) and Ag (2.889Å[50]). Taking the lattice mismatch and the significantly weaker interaction of *h*-BN with the metal surface in comparison to the strong $\sigma$-bonds between N and B atoms[51] into account, a low-strain scenario consists of a $2 \times 2$ supercell of *h*-BN grown on a $\sqrt{3} \times \sqrt{3}R30°$ TM/HM (Pt, Au, Ag), which are commonly found at *h*-BN/HM(111) interfaces[52,53]. From our DFT calculations the lattice mismatch in this scenario is about 3.2% for Pt, 1.3% for Au, and 0.9% for Ag, respectively. We systematically studied all 3*d* transition metal monolayers with three different HM substrates (Pt, Au, Ag), as highlighted in Fig. 1b. We have considered the appropriate magnetic states for the respective 3*d*

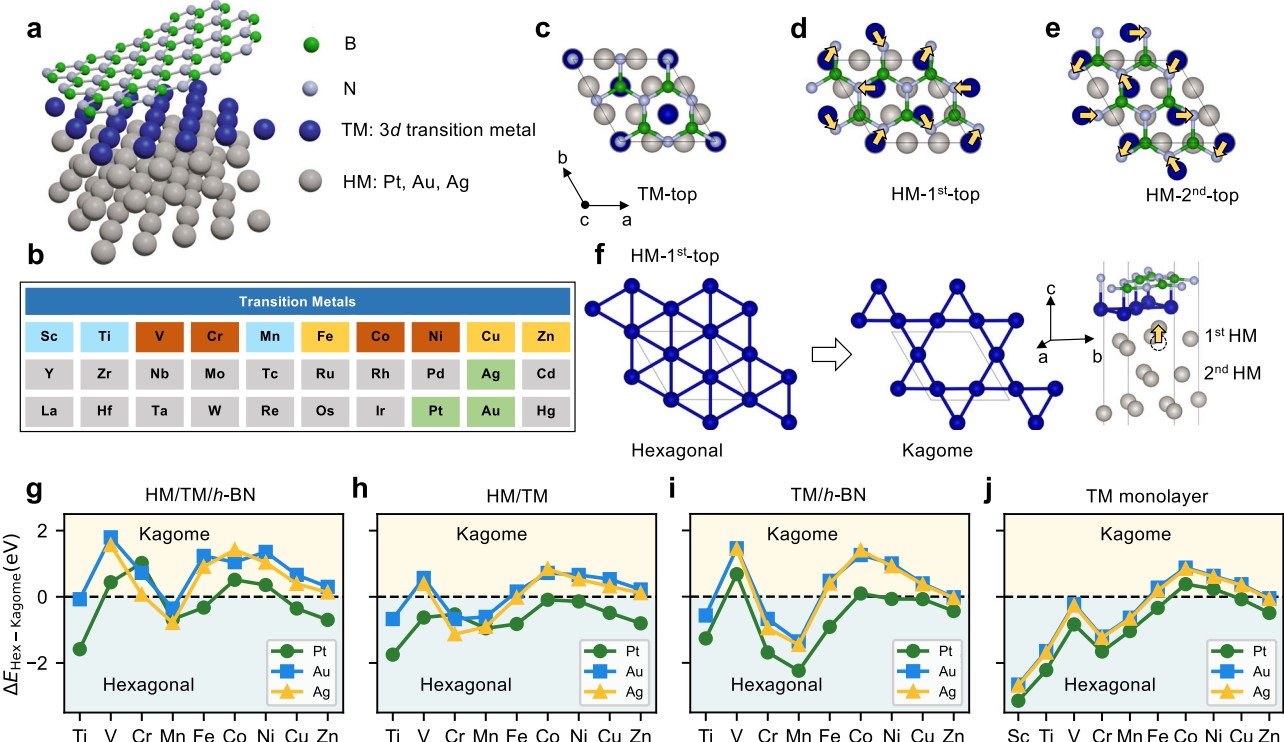

**Fig. 1 | Kagomerization of transition metal monolayers on heavy-metal substrates. a** Schematic representation of the fcc(111) heavy-metal (HM) substrate with an epitaxial 3*d* transition metal (TM) monolayer, capped by *h*-BN. **b** Considered portion of the periodic table. Elements shaded in gray were not investigated. **c–e** Three stacking configurations that have both a N and a B atom directly on top of: **c** the TM atoms (TM-top); **d** the interfacial HM atoms (HM-1st-top); or **e** the sub-interfacial HM atoms (HM-2nd-top). **f** Reshaping of the TM layer in the HM-1st-top structure from an hexagonal to a kagome lattice, shown in top and side views.

**g–j** Energy differences between the hexagonal and the kagome structures of the TM monolayer: **g** for the complete heterostructure; **h** without *h*-BN; **i** without the HM; **j** without *h*-BN and the HM. Positive (negative) values indicate an energetic preference for the kagome (hexagonal) structures. For Sc on all surfaces and for Ti on Ag the initial kagome structure reverted back to the hexagonal one, so no data is shown. The structures in panels **h–j** are not energy-optimized configurations and serve only for comparison purposes.

elements: the ferromagnetic state for Fe, Co and Ni; the triangular antiferromagnetic Néel state for Cr and Mn; the non-magnetic state for the others. We discover that the kagomerization is a rather universal behavior among the explored materials. It can occur spontaneously, even when starting from a non-kagome stacking configuration, and can often be the ground state.

In Fig. 1c−e, we illustrate three different starting configurations for our heterostructures. By performing a DFT relaxation of these structures, we find that the TM atoms always move underneath N atoms, as indicated by the yellow arrows in Fig. 1d,e, except for the TM-top structure, Fig. 1c. This behavior parallels previous findings on Ni(111) and Co(0001) surfaces capped by $h$-BN, for which the favorable binding configuration is also found to be N (B) atoms standing directly on top of the transition metal atom (hollow sites), respectively[54,55]. Due to the electrostatic landscape created by the $h$-BN monolayer, a kagome lattice is obtained from the HM-1$^{st}$-top structure (see Fig. 1f), while the HM-2$^{nd}$-top structure leads to a distorted hexagonal lattice (see Supplementary Fig. S1), and the TM-top structure maintains the ideal hexagonal lattice. In the obtained kagome lattice all N atoms of the unit cell are on top of a TM atom, except for one N that instead attracts one HM atom from the substrate towards the hexagonal hole of the kagome lattice.

Figure 1 g shows the total energy difference between the relaxed hexagonal and kagome structures, where positive (negative) values indicate a kagome (hexagonal) ground state, respectively. We find that V, Cr, Co and Ni monolayers (highlighted in red in Fig. 1b) form a kagome lattice as the ground state on top of the three investigated substrates, while for the Fe, Cu and Zn monolayers this happens only on the Ag and Au surfaces (yellow boxes in Fig. 1b). In some cases (e.g. V and Cr on Pt), although the kagome structure is the ground state, none of the starting configurations shown in Fig. 1c−e relaxes to it, which likely indicates the existence of an energy barrier. Lastly, Sc and Ti are found to never favor the kagome structure. More details about the spontaneity and ground state can be found in Supplementary Fig. S2, while Supplementary Fig. S3 provides more information about various structural parameters.

To identify the mechanisms leading to the kagomerization, we analyze the energy differences plotted in Fig. 1g. First, one immediately recognizes that the likeliness to fabricate a kagome lattice is enhanced when Au and Ag substrates are utilized. Second, the energy differences across the family of transition elements show an M-shape, with a two-peak feature close to the edges of the transition metal series and a minimum around half-filling, i.e., in the middle of the series. This behavior is a signature of a $d$-band-filling effect, which is also used to rationalize the trends of their cohesion[56,57] and surface energies[58,59]. These findings suggest that the tendency of a given transition metal towards forming a kagome structure can be estimated from its placement in the periodic table.

To gain some insights into the role played by $h$-BN, we split the heterostructures by removing either $h$-BN or the metal surface, but keeping all remaining atoms in their prior optimized positions. In Fig. 1h, we plot the total energy differences between the hexagonal and kagome structures when $h$-BN is removed. While for Pt the hexagonal structure is always lower in energy, for Ag and Au the kagome structure still has a lower energy for several elements. This is likely due to combination of the weaker binding between the TM atoms and Ag or Au and the large effective strain from the mismatch between the surface lattice parameter and the equilibrium TM-TM bond lengths. In Fig. 1i, we plot the total energy differences between the hexagonal and kagome structures when instead the HM surface is removed. The trend for whether the kagome or the hexagonal structures have lower energy is broadly similar to what was previously found in the calculations for the complete heterostructures, which points to $h$-BN being indeed the main driver for the kagomerization. As a final check, we consider the energetics of the free-standing monolayer, shown in Fig. 1j.

When the monolayers are strained to match the lattice constants of the absent surfaces the hexagonal structure is favored for most elements, however the kagome structure is preferred for the transition metals going from Fe to Cu. Note that energy-optimized free-standing monolayers always prefer the hexagonal structure, see Supplementary Table S1. This shows that there is an intrinsic component to the kagomerization, which can be enhanced by the combination of surface-driven strain and hybridization with $h$-BN.

To summarize, we found that the kagomerization is a fairly generic behavior for the studied heterostructures. Due to the electrostatic landscape created by $h$-BN, TM atoms interact with $h$-BN and experience an effective attraction towards their closest N atom. However, the TM atoms are also bound to the surface, which can hinder the expected motion towards N. As the interaction of the TM atoms with the surface is weaker for the Ag and Au cases than for the Pt case, so is the kagomerization more energetically favored for the noble metal surfaces. Another factor that contributes to the stabilization of the kagome structure is its shorter nearest-neighbor distance in comparison to the ideal hexagonal structure. This can alleviate some of the strain that the TM atoms are subject to when forced to adopt the lattice parameter of the HM surfaces. Our results provide a practical method to realize the 3$d$ transition metal kagome monolayer and establish a platform for studying intriguing 2D physics.

## Dirac cones, flat bands and van Hove singularities

Having established that TM monolayers can adopt a stable kagome lattice, we first turn to its expected electronic fingerprints, which are Dirac cones and flat bands according to simple nearest-neighbor tight-binding models. We choose FM Co as a representative case, and start by recapitulating the basic phenomenology. In the simplest tight-binding model of a kagome lattice we expect to find three types of van Hove singularities (VHS): one from the flat band and two additional ones from constant-energy contours that pass through the M points in the Brillouin zone. This type of constant-energy contour is also present for a simple tight-binding model of a 2D hexagonal lattice and leads to a single VHS in that case. In both cases the bands crossing at the K-point lead to Dirac cones. In a realistic band structure, the situation is more complex, but we can still use this basic phenomenology for guidance.

In Fig. 2 we compare the band structures of Co monolayers with kagome and hexagonal structures. Other band structure examples are collected in Supplementary Fig. S4. In the free-standing case with the ideal kagome structure, Fig. 2a, we identify several VHS that can be traced to bands which are very flat across extended portions of the Brillouin zone at -0.8, 0.5, 0.6 and 0.8 eV. These VHS singularities seem to be mostly driven by the very flat band dispersion and it is not clear if the M point mechanism is operative and contributing to them. In contrast, for the free-standing case with the hexagonal structure (Fig. 2d) we find a large VHS that is related to the M point around 0.2 eV, with another VHS close by arising from a fairly flat band around Γ. This sets the stage concerning the VHS without effects from hybridization with the surface or interaction with the $h$-BN layer, and the behavior is clearly different from the $A$V$_3$Sb$_5$ systems[33]. In both structures we find the expected Dirac cones at the K point, very close to the Fermi energy.

We next discuss how the surface and the $h$-BN layer modify this picture, and show that VHS from flat bands do indeed survive and are close enough to the Fermi energy to potentially drive interesting physics. For the kagome case, Fig. 2b and c, we still discern flat bands despite the strong hybridization, such as at -0.15 eV in Pt/Co/$h$-BN (Fig. 2b) and at 0.2 eV in Ag/Co/$h$-BN (Fig. 2c), which lead to VHS close to the Fermi energy (indicated by the arrows in those figures). In contrast, flat bands are not observed near the Fermi energy in hexagonal Pt/Co/$h$-BN (Fig. 2e), while some flat bands survive around Γ for Ag/Co/$h$-BN (Fig. 2f). This is a unique feature of the Ag(111) surface

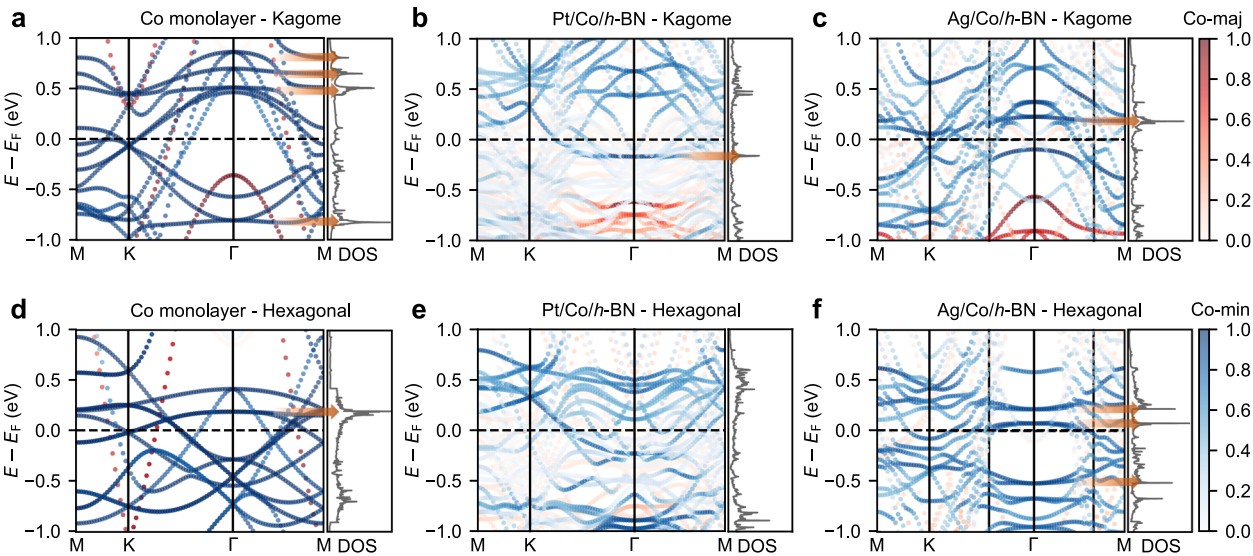

**Fig. 2 | Electronic structure of kagome and hexagonal Co monolayers.** Spin-resolved Co-projected band structures and DOS for: **a** isolated Co monolayer kagome lattice; **b** kagome structure of Pt/Co/*h*-BN; **c** kagome structure of Ag/Co/*h*-BN; **d** isolated Co monolayer hexagonal lattice; **e** hexagonal structure of Pt/Co/*h*-BN; and **f** hexagonal structure of Ag/Co/*h*-BN. The arrows indicate some van Hove singularities in the band structures and DOS.

that contains no electronic states in a broad energy region around Γ, preventing hybridization with the Co bands. The Dirac crossings are shifted to above the Fermi energy, and in the Ag case are not as well isolated from other bands as in the Pt case.

We conclude that for this Co example most of the flat bands are not robust against hybridization with the surface electrons, except for those in the kagome lattice near the Fermi energy. More generally, the fundamental electronic aspects expected for a kagome lattice are indeed found in the discussed kagomerized monolayers, which hence offer a new platform for the investigation of potential correlated topological phases.

## Impact of kagomerization on magnetic properties and magnetic interactions

We now discuss the kagome-related changes to the TM magnetism, concentrating on FM Fe, Co and Ni for simplicity. We first compare the spin moments for the HM/TM (hexagonal, labeled w/o-BN-Hex) and for the HM/TM/*h*-BN (kagome, labeled w/-BN-Kagome) structures. The overall trend is for a reduction when going from the hexagonal to the kagome structure (for instance, the spin moment of Fe on Au reduces from 3.09 $\mu_B$ to 2.78 $\mu_B$), and we gather all the values in Supplementary Table S2. The question arises as to whether this reduction is driven mostly by the structural rearrangement (and the attending shortening of the TM nearest-neighbor distances), by the hybridization with *h*-BN, or by a combination of both. As this will also be pertinent for the magnetic interactions, we devise two auxiliary structures to acquire additional data. The w/-BN-Hex structure is obtained from the HM/TM structure by introducing the *h*-BN overlayer while maintaining the HM layer in the hexagonal arrangement, and the w/o-BN-Kagome structure by removing the *h*-BN overlayer from the full kagome structure. These four structures are illustrated in Supplementary Fig. S5. We computed the pairwise magnetic interactions, including Heisenberg exchange (*J*) and DMI (**D**), for all structures (see Methods section).

We select the Pt/Co system to illustrate the changes to the magnetic interactions due to the kagomerization before presenting a systematic overview. Figure 3a shows the spatial distribution of *J* for the epitaxial Co on Pt structure (w/o-BN-Hex) with $C_{3v}$ symmetry. Each circle represents a Co atom and is colored as a function of the *J* values with respect to the central Co (gray circle), which is taken as reference. The value of *J* is given by the color scale, with red (blue) for ferromagnetic (antiferromagnetic) coupling. Figure 3b depicts the results

obtained for the w/o-BN-Kagome structure. The nature of the interactions as a function of the distance is similar to the hexagonal case. The results for the structures that include the *h*-BN overlayer are shown in Fig. 3c–d, but to make a comparison we also compute the differences to the results obtained without the overlayer, for the same arrangement of Co atoms. This is presented in Fig. 3e–f, with the purple (green) color revealing an increase (decrease) of *J* after capping with *h*-BN. The largest absolute change in the interactions takes place for the nearest-neighbor pairs, as these are the strongest interactions. For example, the nearest-neighbor interactions in the w/-BN-Kagome structure are 4 meV larger than in the w/o-BN-Kagome structure, a relative change of 14%. However, the largest relative changes occur for larger distances for which the interactions are small in magnitude.

An alternative way of quantifying the differences in the magnetic interactions between the different structures is by computing the respective micromagnetic parameters, defined in the Methods. As the sums run over all the magnetic atoms, the micromagnetic quantities have the symmetry of the point group, which is higher than the symmetry of the pairwise interactions shown in Fig. 3b–d, and results in only two independent parameters (one for the stiffness and one for the spiralization). The long-range part of the pairwise interactions makes a very important contribution to the micromagnetic parameters (see Supplementary Note 5 and Supplementary Fig. S6a), so it cannot be neglected. Figure 3g shows the independent parameter $\mathcal{A}_{xx}$ for the stiffness tensor for the considered systems (by symmetry, $\mathcal{A}_{yy} = \mathcal{A}_{xx}$ and all others are zero). Looking first at the results for the Pt/Co system and taking the value obtained for the w/o-BN-Hex structure as reference, we find a modest change in $\mathcal{A}_{xx}$ except for the realistic w/-BN-Kagome structure for which there is a 39% reduction. This is explained by the changes in the long-range part of the Heisenberg exchange that were mapped in Fig. 3f. This strong reduction cannot be explained either by the formation of the kagome structure or by the hybridization with *h*-BN, and is clearly a combined effect of the two. The nature of the surface also plays an important role, as this strong change in $\mathcal{A}_{xx}$ is not seen for the Au/Co and Ag/Co systems. This synergy between structural rearrangement and hybridization is even more prominent for the Ni systems, for which we found an almost complete suppression of $\mathcal{A}_{xx}$ for Au/Ni and Ag/Ni due to a strong enhancement in antiferromagnetic character of the long-range interactions. Lastly, the case of Ag/Fe shows that a significant strengthening of $\mathcal{A}_{xx}$ due to hybridization with *h*-BN is also possible.

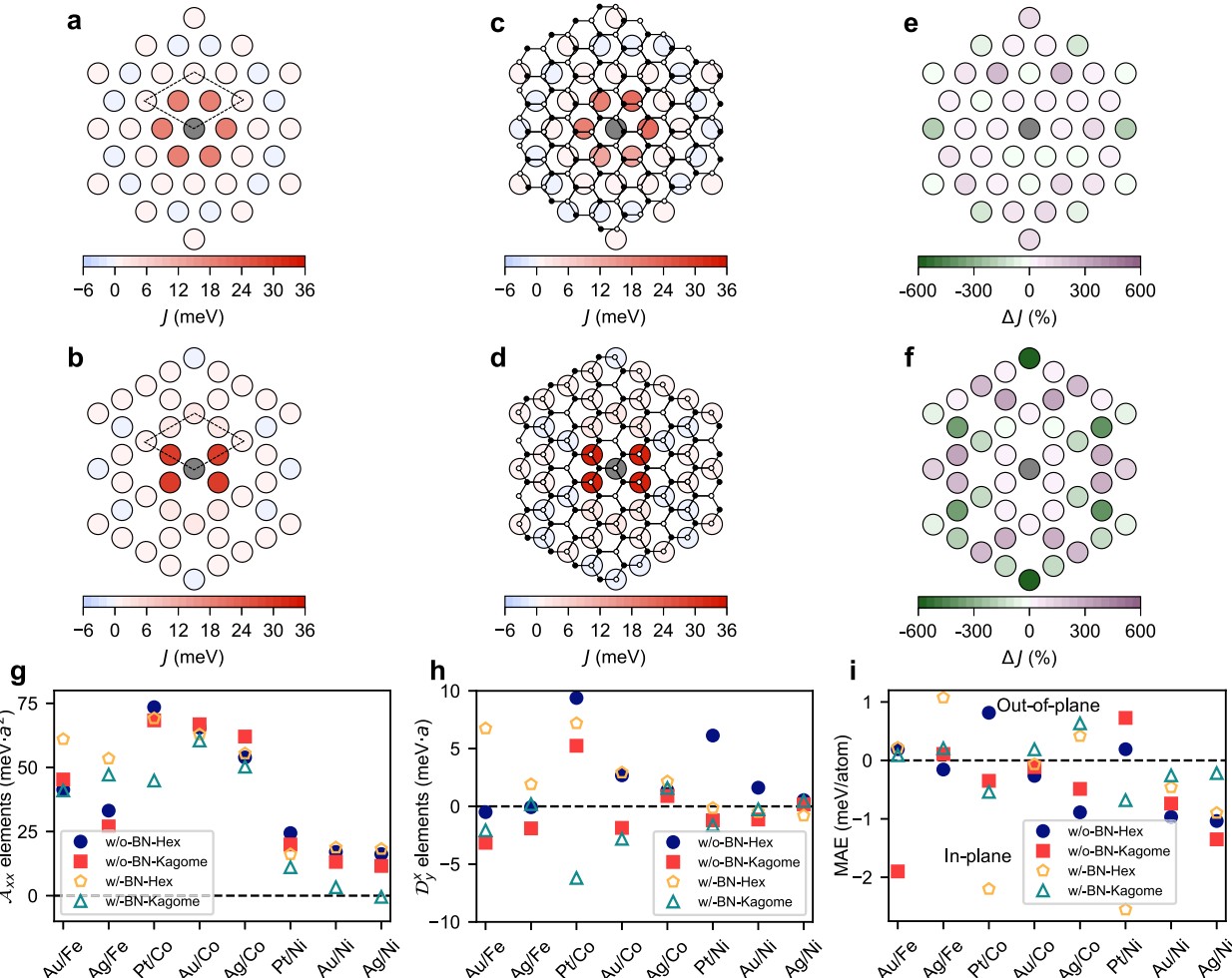

**Fig. 3 | Impact of kagomerization on the magnetic interactions and magnetic anisotropy energy (MAE). a-d** Spatial distribution of the Heisenberg exchange interactions for the four considered structures in the Pt/Co system. The unit cell is shown by dashed black lines and contains three Co atoms in all cases. The Co atom shaded in gray is taken as the reference atom and the color of the surrounding circles indicates the value of $J$ for the corresponding pair. The positive (negative) values correspond to ferromagnetic (antiferromagnetic) coupling. Shown are the **a** w/o-BN-Hex, **b** w/o-BN-Kagome, **c** w/-BN-Hex, and **d** w/-BN-Kagome structures. The small white and black circles represent N and B atoms, respectively. **e-f** Relative differences in $J$ values of Pt/Co system for the structures with and without $h$-BN. **e** corresponds to **c** minus **a**. **f** corresponds to **d** minus **b**. Micromagnetic parameters: **g** Exchange stiffness tensor elements $\mathcal{A}_{xx}$; **h** DMI spiralization tensor elements $\mathcal{D}_y^x$; **i** MAE per magnetic atom. Positive (negative) values correspond to out-of-plane (in-plane) anisotropy.

The properties of the pairwise DMI are nicely condensed in their micromagnetic counterpart, the DMI spiralization tensor (see Methods), which thanks to the point group symmetry only has one independent parameter $\mathcal{D}_y^x$ ($\mathcal{D}_x^y = -\mathcal{D}_y^x$ and all other elements are zero). Figure 3h shows the values of $\mathcal{D}_y^x$ for all considered structures. Pt has a strong spin-orbit coupling and hybridizes strongly with the magnetic atoms, which leads to the strong DMI found for Co and Ni in the w/o-BN-Hex structure, while the weaker hybridization for Au and the weaker spin-orbit coupling of Ag explain the weaker DMI values found in those systems. In contrast, spin-orbit coupling is negligible for $h$-BN, so its strong impact on the DMI values is driven both by the structural rearrangement of the magnetic atoms and by modifications to their hybridization with the surface. We first demonstrate that the DMI is quite susceptible to the structural rearrangement from hexagonal to kagome. Comparing $\mathcal{D}_y^x$ in the w/o-BN-Hex and w/o-BN-Kagome structures, we even find that the DMI chirality is reversed in the Au/Co, Pt/Ni and Au/Ni systems. The modifications to the DMI are even stronger for the Pt/Co system, but only once $h$-BN is accounted for. The hybridization with $h$-BN can even induce sizeable DMI for systems where it was originally negligible, such as Au/Fe and Ag/Fe.

The final aspect to consider is the modification of the magnetic anisotropy energy (MAE) by $h$-BN. The MAE for all the structures is shown in Fig. 3i. Positive values indicate an out-of-plane and negative values an in-plane anisotropy, respectively. Going from the w/o-BN-Hex structure to the w/-BN-Kagome one can even lead to a change in the nature of the MAE, from out-of-plane to in-plane for Pt/Co and Pt/Ni and in reverse for Ag/Fe, Au/Co, Ag/Co (and almost for Ag/Ni). Overall, we conclude that the kagomerization of the TM layer driven or stabilized by $h$-BN leads to quantitative and often qualitative changes in the various considered magnetic interactions, and so we also expect strong changes to the magnetic states found for each system.

## Spin textures and their evolution in a magnetic field

Figure 4 summarizes the magnetic states found by atomistic spin dynamics simulations utilizing the full set of magnetic interactions obtained for the two realistic structures, w/o-BN-Hex and w/-BN-Kagome. Without $h$-BN we find ferromagnetic states for most systems, which are in-plane or out-of-plane according to the respective MAE, with the exception of the Pt systems for which the DMI is strong enough to enable metastable skyrmionic states. $h$-BN changes the

| Hexagonal lattice | | | Kagome lattice | | |
| --- | --- | --- | --- | --- | --- |
| Au/Fe | Ferromagnetic | | Au/Fe/h-BN | Skyrmions | |
| Ag/Fe | Ferromagnetic | | Ag/Fe/h-BN | Ferromagnetic | |
| Pt/Co | Skyrmions | | Pt/Co/h-BN | Bimerons | |
| Au/Co | Ferromagnetic | | Au/Co/h-BN | Ferromagnetic | |
| Ag/Co | Ferromagnetic | | Ag/Co/h-BN | Ferromagnetic | |
| Pt/Ni | Skyrmions | | Pt/Ni/h-BN | Ferromagnetic | |
| Au/Ni | Ferromagnetic | | Au/Ni/h-BN | Bimerons | |
| Ag/Ni | Ferromagnetic | | Ag/Ni/h-BN | Bimerons & spirals | |

**Fig. 4 | Magnetic states for various heterostructures.** Magnetic states found for the magnetic monolayers on heavy-metal surfaces and their modification by $h$-BN. These were obtained from atomistic spin dynamic simulations employing the magnetic interactions for the two realistic structures, w/o-BN-Hex and w/-BN-Kagome.

magnetic states for different systems in different ways. In the Pt/Co case, skyrmions turn into bimerons due to the change of sign of the MAE, while for the Pt/Ni case an in-plane ferromagnetic state is stabilized by a combination of reduced DMI and strengthened in-plane MAE. (We remark that a bimeron can also be referred to as an asymmetric skyrmion[60] or in-plane skyrmion[61,62]). We also found qualitative changes in the magnetic state for three other systems. Skyrmions appear for Au/Fe due to the enhanced DMI, while for Au/Ni and Ag/Ni bimerons appears due to a weakening of the MAE and a dramatic suppression of the exchange stiffness. In the Ag/Ni system this suppression is strong enough to destabilize the ferromagnetic state itself in favor of a spin spiral. The spatial dimensions of these topological spin textures in our kagome structures range from about 2 to 13 nm, and skyrmions with similar sizes have been found in related heterostructures[63,64].

Next we investigate the response of the various spin textures to an applied magnetic field. The considered zero-field spin textures are presented in Fig. 5a−c. For Pt/Co/$h$-BN (Fig. 5a) and Au/Ni/$h$-BN (Fig. 5b), bimerons prefer to be in the head-to-head configuration and connect with each other. To compare with the more familiar skyrmion picture, we uniformly rotate the magnetization by 90° so that its ferromagnetic component is along the $z$-axis, obtaining the distorted skyrmion state shown in the insets. Note that the uniform rotation applied to all the spins is only used for visualization purposes, in order to aid our subsequent analysis. This establishes that the bimerons themselves are asymmetric. The situation is more complex for Ag/Ni/$h$-BN (Fig. 5c), where we find merons and antimerons trapped in spin spirals. The total number of merons or antimerons in the system is always even, so we refer to them as bimerons.

The obtained bimerons are stabilized by the combination of interface-induced DMI and in-plane anisotropy, which leads to their discussed asymmetry, and will also lead to different responses to external fields applied along the $+z$ or $−z$ directions. The evolution of the spin textures with the applied field is summarized in Fig. 5d−f. This evolution can be divided into three regions indicated by different colors. Bimerons exist in the yellow region, and for increasing magnitude of the applied field they deform and become progressively more similar to skyrmions or antiskyrmions. Skyrmions exist in the pink region, which only appears for a given sign of the magnetic field but not for the opposite one in Fig. 5d−e. This is due to the asymmetry of the bimerons, which causes them to annihilate in the ferromagnetic state for the unfavorable sign of the magnetic field. The situation is qualitatively distinct for Ag/Ni/$h$-BN (Fig. 5c) as the bimerons exist in

a spin spiral background. This introduces a compensation in the possible types of asymmetry for the bimerons, so that some can always evolve into skyrmions irrespective of the sign of the applied field. One could then characterize the yellow region in Fig. 5f as containing bimerons which are in a sense precursors of skyrmions or antiskyrmions which would arise in the applied field, the skyrmions surviving into the pink regions and the antiskyrmions annihilating into the ferromagnetic state.

## Discussion

We identified through first-principles simulations a spontaneous rearrangement of transition metal atoms on a heavy-metal surface catalyzed by an $h$-BN overlayer − a process that we named kagomerization. The selected heavy-metal surfaces (Pt, Au and Ag) were chosen due to the small lattice mismatch between a $2 \times 2$ supercell of $h$-BN and a $\sqrt{3} \times \sqrt{3}R30°$ surface unit cell. We anticipate that kagomerization can also take place on other surfaces. We believe that the proposed structures are experimentally realizable, given that magnetic monolayers on heavy-metal substrates have been studied before[65,66], and there are works reporting Co films on Pt or Au covered with $h$-BN[43]. In fact, there is a recent experimental report of a monolayer Ni kagome lattice forming on Pb(111), which was found to be non-magnetic[67]. There are also proposals for obtaining two-dimensional magnetic kagome lattices through mechanical exfoliation of van der Waals materials, for instance $Nb_3X_8$ (X = Cl, Br, and I)[68–71]. Here our focus is on more conventional heterostructures that are routinely grown for spintronics experiments, concentrating on material combinations which preserve their magnetism and have a higher likelihood of being experimentally realized. The band structure of the kagomerized structures shows the expected flat band and Dirac cone features, which should provide opportunities to study the interplay between geometry, topology, and magnetism.

The central aspect of our study is the systematic study of how the kagomerization modifies the magnetic properties of the transition metal monolayers. We selected three ferromagnetic elements, Fe, Co and Ni, and adopted a two-pronged approach based on extracting all the relevant magnetic interactions from first-principles calculations combined with an exploration of the magnetic states through atomistic spin dynamics. Our analysis shows that $h$-BN often has a strong quantitative and even qualitative effect on the magnetic interactions, challenging its assumed role as a passive capping layer. Both the spin-orbit-driven magnetic anisotropy energy and Dzyaloshinskii-Moriya interactions as well as the typically much

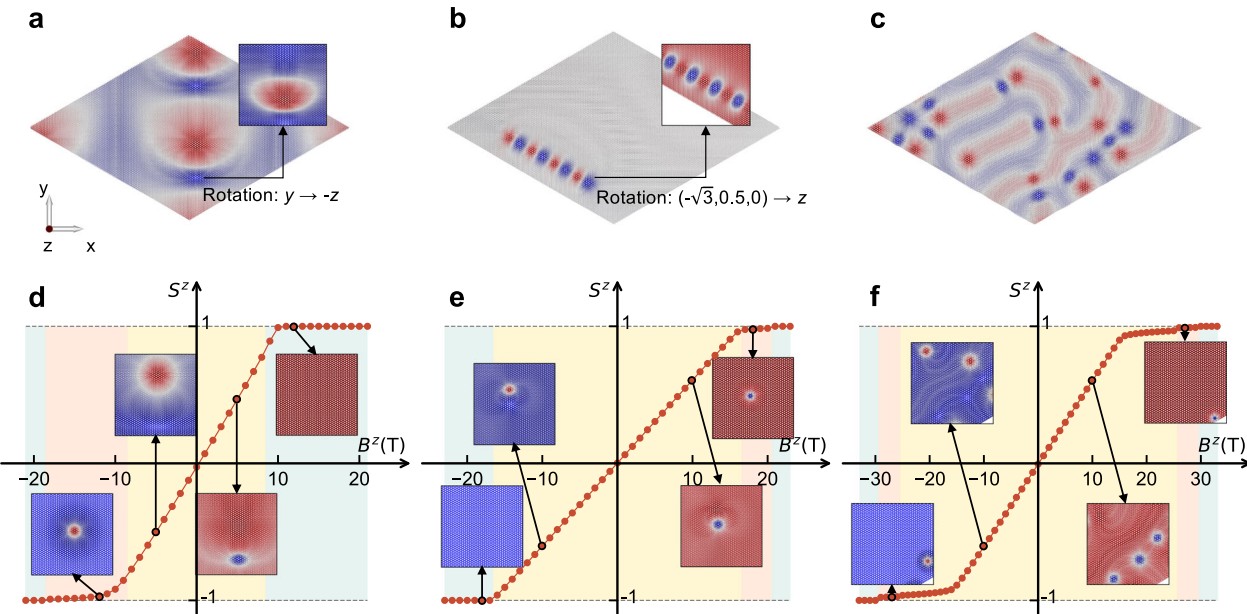

**Fig. 5 | Manipulating spin textures with a magnetic field. a–c** Spin textures in zero external magnetic field: **a** Bimerons in Pt/Co/*h*-BN; **b** Bimeron chain in Au/Ni/*h*-BN; **c** Spirals with trapped bimerons in Ag/Ni/*h*-BN. **d–f** Evolution of spin textures with a magnetic field, traced with the total *z* component of the magnetization normalized by the number of sites: **d** Pt/Co/*h*-BN; **e** Au/Ni/*h*-BN; **f** Ag/Ni/*h*-BN. We identify three regions with different colors: yellow, pink, and blue blocks indicate the regions where bimeron, skyrmion, and uniform ferromagnetism occur, respectively. The insets show examples of the corresponding magnetic states. The spins are colored as follows: red is + *z*, blue is − *z*, and gray is in-plane.

stronger Heisenberg exchange interactions can be strongly modified by the kagomerization. This enables switching between different magnetic states, such as from skyrmions to bimerons, or from ferromagnetic to spiraling states, by the addition of an *h*-BN overlayer. Furthermore, we unveiled the asymmetric response of the bimerons under external magnetic fields applied along the positive and negative *z* directions.

Our prediction of kagomerization of transition metal layers by *h*-BN and the surprisingly strong impact on their magnetic properties opens exciting perspectives for spintronics. Besides exploring the properties of the homogeneous heterostructures and their tunability with an applied magnetic field, additional prospects are opened by creating areas with and without the *h*-BN overlayer, for instance by lithography. These different areas can host very different magnetic states, and could act as spintronic circuit elements for skyrmion or bimeron injection or annihilation, among other possibilities. The combination of fundamental physics with prospects for innovative device concepts will make this type of heterostructure very appealing for future investigations.

## Methods
### First-principles calculations
Optimization of the considered structures was performed using the density functional theory (DFT) approach implemented in the Quantum Espresso computational package[72], including the van der Waals correction (DFT-D3[73]). We employed the projector augmented wave pseudopotentials from the pslibrary[74], with the generalized gradient approximation of Perdew, Burke and Ernzerhof (PBE)[75] as the exchange and correlation functional. Convergence tests led to a plane-wave energy cut-off of 90 Ry and an $18 \times 18 \times 1$ k-point grid. We optimize the lattice constant of the HM substrates and use those values for the heterostructures with TM layers and with or without *h*-BN.

The magnetic properties and magnetic interactions were computed using the all-electron full-potential scalar-relativistic Korringa-Kohn-Rostoker (KKR) Green function method[76–78] including spin-orbit coupling self-consistently as implemented in the JuKKR computational package. The angular momentum expansion of the Green function was truncated at $\ell_{\max} = 3$ with a k-mesh of $18 \times 18 \times 1$ points. The energy integrations were performed including a Fermi-Dirac smearing of 502.78 K, and the local spin-density approximation (LSDA) was employed[77]. The Heisenberg exchange interactions and DM vectors were extracted using the infinitesimal rotation method[79,80] with a finer k-mesh of $80 \times 80 \times 1$ and real-space cutoff radii of $R = 14a$ for Fe and $R = 8a$ for Co and Ni (*a* is the lattice parameter of the surface unit cell). The combination of PBE for geometry optimization and LSDA for magnetic properties has been widely used and those works achieve good agreement between theory and experiment[81–86]. Our test calculations show that different functionals do not affect the occurrence of kagomerization and have no significant impact on the magnetic interactions (see Supplementary Note 6).

### Magnetic interactions and atomistic spin dynamics
We consider a classical extended Heisenberg Hamiltonian including Heisenberg exchange coupling (*J*), DMI (**D**), the magnetic anisotropy energy (*K*), and Zeeman term (**B**). All parameters were obtained from first-principles calculation. The spin-lattice model reads as follows:

$$E = -\sum_{i,j} J_{ij}\, \mathbf{S}_i \cdot \mathbf{S}_j - \sum_{i,j} \mathbf{D}_{ij} \cdot \left( \mathbf{S}_i \times \mathbf{S}_j \right) - \sum_i \mathbf{B} \cdot \mathbf{S}_i - \sum_i K_i (S_i^z)^2 . \quad (1)$$

where $\mathbf{S}_i$ and $\mathbf{S}_j$ are the magnetic moments of unit length at position $\mathbf{R}_i$ and $\mathbf{R}_j$ respectively.

We will transform the obtained pairwise magnetic interactions into the parameters of the following micromagnetic energy functional:

$$E_{\text{micro}} = \frac{1}{V_0} \int d\mathbf{r} \left[ \frac{1}{2} \sum_{\alpha,\beta} \mathcal{A}_{\alpha\beta}\, \frac{\partial \mathbf{m}(\mathbf{r})}{\partial r^\alpha} \cdot \frac{\partial \mathbf{m}(\mathbf{r})}{\partial r^\beta} - \sum_{\alpha,\mu} \mathcal{D}_\alpha^\mu\, \mathcal{L}_\alpha^\mu(\mathbf{r}) - \mathbf{B} \cdot \mathbf{m}(\mathbf{r}) - K (m^z(\mathbf{r}))^2 \right],$$

$$(2)$$

where

$$\mathcal{A}_{\alpha\beta} = \frac{1}{N} \sum_{i,j} J_{ij} (R_j^\alpha - R_i^\alpha)(R_j^\beta - R_i^\beta) \tag{3}$$

is the exchange stiffness tensor, and

$$\mathcal{D}_\alpha^\mu = \frac{1}{N} \sum_{i,j} D_{ij}^\mu \left( R_j^\alpha - R_i^\alpha \right) \tag{4}$$

is the DMI spiralization, and

$$\mathcal{L}_\alpha^\mu(\mathbf{r}) = \sum_{\nu,\eta} \varepsilon_{\mu\nu\eta} \, m^\nu(\mathbf{r}) \, \frac{\partial m^\eta(\mathbf{r})}{\partial r^\alpha} \tag{5}$$

are the Lifshitz invariants. Here $\alpha, \beta, \gamma$ label the Cartesian components of the position vector of each atom **R** and of the continuum position **r** for the magnetization vector field, and $\mu, \nu, \eta = x, y, z$ indicate the three components of the magnetization direction, with $\varepsilon_{\mu\nu\eta}$ the Levi-Civita symbol. $V_0$ is the volume per magnetic site and the sums run over all the $N$ magnetic atoms for which the interactions are defined.

Complex magnetic states are explored through atomistic spin dynamic simulations using the Landau-Lifshitz-equation (LLG) as implemented in the Spirit code are performed using the Hamiltonian from Eq. (1). We employed boundary conditions to model the extended two-dimensional system on a lattice with 99 × 99 unit cells, each containing 3 spins.

## Data availability
The data needed to evaluate the conclusions in the paper are present in the paper and the Supplementary Information.

## Code availability
The codes employed for the simulations described within this work are open-source and can be obtained from the respective websites and/or repositories. Quantum Espresso can be found at[88], and the Jülich-developed codes JuKKR and Spirit can be found at[89] and[90], respectively.

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

## Acknowledgements
S.L. acknowledges fruitful discussions with Salim M. Cherif. This work was supported by the China Scholarship Council program (H.Z.) and Outstanding Research Project of Shen Yuan Honors College, BUAA (230121102, H.Z.), the Priority Programmes SPP 2244 "2D Materials Physics of van der Waals heterobilayer" (Project LO 1659/7-1, S.L.), SPP 2137 "Skyrmionics" (Project LO 1659/8-1, S.L.) of the Deutsche Forschungsgemeinschaft (DFG), National Key Research and Development Program of China (2022YFB4400200, W.Z.), National Natural Science Foundation of China (92164206, W.Z.; 52121001, W.Z.), the New Cornerstone Science Foundation through the XPLORER PRIZE (W.Z.), CoSeC and the Computational Science Centre for Research Communities (CCP9, M.d.S.D.). Simulations were performed with computing resources granted by RWTH Aachen University under project p0020362 and JARA on the supercomputer JURECA[87] at Forschungszentrum Jülich.

## Author contributions
S.L. initiated, designed and supervised the project. H.Z. performed the simulations with support and supervision from M.d.S.D and S.L. H.Z., M.d.S.D., Y.Z., W.Z., and S.L. discussed the results. H.Z., M.d.S.D., and S.L. wrote the manuscript to which all co-authors contributed.

## Funding

## Competing interests
The authors declare no competing interests.
