## [Peer Review File · Nature Communications]

Kagomerization of transition metal monolayers induced by two-dimensional hexagonal boron nitrideREVIEWER COMMENTS

Reviewer #1 (Remarks to the Author):

The manuscript presents a theoretical study of structural, electronic and magnetic properties of transition metal (TM) atoms supported on a fcc(111) heavy-metal (HM) surface with an hexagonal boron nitride (h-BN) overlayer. The authors found that a large rearrangement into two-dimensional kagome lattice of the TM atoms occurs often in this type of heterostructures, along with characteristics of Dirac cones and flat bands. They also revealed that this kagomerization process can modify the magnetic properties and magnetic interactions of the heterostructures, possibly leading to formation of some specific topological spin textures, such as skyrmions and bimerons, adjusted by the magnetic field. The results are interesting and important, especially that the h-BN overlayer plays roles beyond encapsulation, and their analyses on the formation mechanisms of the two-dimensional kagome lattice and related topological spin textures are also impressive and sound credible. This study will be considerably beneficial for the future explorations in the spintronics and related physics. The manuscript deserves publication in a high-impact journal.

A main problem is lack of feasibility and stability analysis on the whole heterostructures consisting of the h-BN overlayer, TM atoms and fcc(111) HM surface. Of course the authors discussed low lattice mismatch between the h-BN overlayer and the HM surface in the selected supercell. However, introducing the TM atoms may bring more uncertainties, such as coverage control of the TM atoms on fcc(111) HM surface and their possible clustering. The speculations about promising experimental realization of this type of heterostructures are also encouraged.

The authors are expected to evaluate influence of different functionals used in their calculations on the obtained results, especially topological spin textures.

The statement of rotating the magnetization makes me confused. Does the authors mean rotating only the magnetization directions of every atoms, not others? This will result in wrong ground states. If the plane of two-dimensional kagome lattice is also rotated simultaneously, the distorted skyrmion and the asymmetric bimerons stated by them become incomprehensible.

“Au/Co/h-BN” on line 254 should be “Au/Ni/h-BN”.

Reviewer #2 (Remarks to the Author):

In the manuscript under consideration here, the authors employ first-principles calculations to explore the impact of h-BN capping on transition metal atoms deposited on heavy metals surfaces. Contrary to conventional wisdom, which wants h-BN merely as an inert capping layer, this study unveils the substantial influence that h-BN has on structural, electronic, and magnetic properties.

More specifically, the authors reveal that h-BN serves as a driving force, inducing a structural kagomerization within transition metal monolayers. This process yields electronic dispersions characterized by prototypical kagome features, including Dirac states and flat bands. Furthermore, h-BN induces a significant rearrangement of magnetic properties,

evident in modifications to magnetic exchange interactions, the Dzyaloshinskii-Moriya interaction, and magnetic anisotropy energy. These modifications ultimately give rise to transformed magnetic states, leading to the emergence of skyrmions and bimerons, which are absent in the absence of h-BN capping.

The study is conducted with meticulous attention to detail and follows a systematic approach. The main message that h-BN's behavior extends beyond a simple capping role is both intriguing and valuable for the scientific community. However, before recommending publication, it is important for the authors to address the following questions.

1) I find it unclear whether the atomic relaxations were carried out in the paramagnetic state or if the authors also considered magnetism. If the calculations were conducted without incorporating magnetism, I expect significant modifications in the hexagonal versus kagome energy difference of Figure 1.

2) Figure 2 illustrates a strong hybridization between the kagome-like bands and the heavy metal bands. I am skeptical about the authors' claim that these kagomerized systems provide a new platform for investigating potential correlated topological phases. Furthermore, the kagome lattice is characterized by van Hove singularities exhibiting both pure and mixed sublattice characters, considered pivotal in generating exotic collective orderings observed in kagome metals like AV₃Sb₅. It remains unclear how the twofold nature of van Hove singularities changes upon deposition on the heavy metal surface and h-BN capping.

3) Are the relatively flat bands observed in Figure 2 panels b) and c) absent in the hexagonal configurations? In other words, are the flat bands a consequence of the kagomerization?

4) What is the spatial extent of the skyrmions obtained through atomistic spin dynamic simulations? How does this extension compare to the typical dimensions observed on established platforms for skyrmions?

5) The majority of the results are presented for Ni, Fe, and Co. However, as indicated in Figure 1, the kagome structure is already the most stable for these transition metals even in the free-standing case, rendering kagomerization unnecessary. Notably, no results are provided for transition metals that exclusively favor the kagome lattice through kagomerization. I perceive this as a potential limitation in the study.

Reviewer #3 (Remarks to the Author):

Kagome lattices are emerging as an exciting platform for the rich physics, including magnetism, charge density wave (CDW), topology, and superconductivity. Thus exploring new materials with Kagome lattice structure is a very interesting research.

In this manuscript, based on first-principle calculation, the author search the possible two-dimensional kagome lattice in monolayers of transition metals, such kind of materials is still not exist to my knowledge. They find that heterostructures based on h-BN can result in kagomerization.

I think this work is quite interesting and can be accepted by NC.

Reply to the Reviewers

In the following, the original comments from the reviewers are reproduced together with our responses in blue and the changes made to the manuscript in red.

Reviewer #1

Reviewer #1: The manuscript presents a theoretical study of structural, electronic and magnetic properties of transition metal (TM) atoms supported on a fcc(111) heavy-metal (HM) surface with an hexagonal boron nitride (h-BN) overlayer. The authors found that a large rearrangement into two-dimensional kagome lattice of the TM atoms occurs often in this type of heterostructures, along with characteristics of Dirac cones and flat bands. They also revealed that this kagomerization process can modify the magnetic properties and magnetic interactions of the heterostructures, possibly leading to formation of some specific topological spin textures, such as skyrmions and bimerons, adjusted by the magnetic field. The results are interesting and important, especially that the h-BN overlayer plays roles beyond encapsulation, and their analyses on the formation mechanisms of the two-dimensional kagome lattice and related topological spin textures are also impressive and sound credible. This study will be considerably beneficial for the future explorations in the spintronics and related physics. The manuscript deserves publication in a high-impact journal.

Authors: We are grateful to the reviewer for carefully reading our manuscript and for the valuable comments and positive assessment of our work. In order to address the review's concerns, we have performed a detailed analysis as described below. We hope the reviewer will be satisfied with the revised manuscript as well as our response.

Reviewer #1– Comment 1: A main problem is lack of feasibility and stability analysis on the whole heterostructures consisting of the h-BN overlayer, TM atoms and fcc(111) HM surface. Of course the authors discussed low lattice mismatch between the h-BN overlayer and the HM surface in the selected supercell. However, introducing the TM atoms may bring more uncertainties, such as coverage control of the TM atoms on fcc(111) HM surface and their possible clustering. The speculations about promising experimental realization of this type of heterostructures are also encouraged.

Authors: We thank the reviewer for this pertinent comment. Based on the existing experimental literature, it is feasible to have such a sandwich structure involving a two-dimensional (2D) van der Waals (vdW) monolayer and controlled-thickness TM thin films with an HM substrate [1–5]. For the particular case of *h*-BN, successful deposition by electron-beam evaporation onto Co/HM has been achieved, though with a thicker layer of Co rather than a monolayer [5]. Experimental studies have also demonstrated the deposition of graphene onto monolayers of Co or Fe on an Ir substrate [2,3]. We anticipate that by careful control of experimental growth conditions it is possible to achieve large areas of uniform coverage instead of patches or island growth. The kagomerization that we are proposing is a theoretically robust process.

In this light, we have added the following to the Discussion section of the main manuscript: **We believe that the proposed structures are experimentally realizable, given that magnetic monolayers on heavy-metal substrates have been studied before [2, 3], and there are works reporting Co films on Pt or Au covered with *h*-BN [5].**

Reviewer #1– Comment 2: The authors are expected to evaluate influence of different functionals used in their calculations on the obtained results, especially topological spin textures.

Authors: We highlight that we are using a well-established methodology for the type of systems that we are addressing. There are two aspects which are difficult to reconcile with the same functional: structural properties and magnetic properties. For the structural properties, the generalized gradient approximation of Perdew, Burke and Ernzerhof (GGA-PBE) is known to give very good results, and so the considered structures were relaxed using the Quantum Espresso package with this functional. For the magnetic properties and magnetic interactions, we used the local spin-density approximation (LSDA) and the all-electron JuKKR computational package. This combination of GGA-PBE and LSDA functionals has been widely used before and good agreement has been found between theory and experiment [6–11]. We have also checked the relevance of accounting for van der Waals interactions in the optimized structures. These interactions are more relevant for physisorption (large separation between *h*-BN and the TM layer) than for chemisorption (small separation between those two layers). Regarding magnetic properties, it is known that PBE tends to overestimate the magnetic moments, which affects the magnetic interactions and magnetic anisotropy energy (MAE). We tested the influence of all these aspects for our systems.

In Table R1, we compared the result in the manuscript (PBE+vdW) with those using LSDA, considering the energy differences $\Delta E_{\text{Hex-Kagome}}$ and geometries for three typical examples: Pt/Co/*h*-BN, Au/Ni/*h*-BN and Ag/Ni/*h*-BN. Our results consistently demonstrate that kagome structures are more stable than hexagonal structures across all the functionals we examined, even if in some cases different functionals switch between chemisorption and physisorption (e.g., hexagonal structure of Pt/Co/*h*-BN). Turning to the magnetic properties, we computed the Heisenberg exchange interaction, Dzyaloshinskii-Moriya interaction (DMI) and MAE of Pt/Co to compare the influence of different functionals on the obtained results in JuKKR code. In Fig R1, we present the micromagnetic parameters (exchange-stiffness-tensor element \mathcal{A}_{xx} and DMI-spiralization-tensor element \mathcal{D}_y^x) as a function of the cut-off radius, and we find only minor discrepancies between the two functionals. However, for the MAE we obtained 0.82 meV/Co with LSDA and 0.59 meV/Co with PBE, and this influences the

Table R1: Energy differences $\Delta E_{\text{Hex-Kagome}}$ (eV) between the hexagonal and the kagome structures of the TM monolayer for the complete structure and geometries (\AA) of both kagome and hexagonal structures: vertical BN-TM ($z_{\text{BN-TM}}$) and vertical TM-HM ($z_{\text{TM-HM}}$) distances.

Pt/Co/ h -BN		Kagome		Hexagonal	
		$\Delta E_{\text{Hex-Kagome}}$	$z_{\text{BN-Co}}$	$z_{\text{Pt-Co}}$	$z_{\text{BN-Co}}$
PBE+vdW	0.51	2.19	2.11	3.13	1.96
LSDA+vdW	1.17	2.02	2.02	2.02	1.86
LSDA	1.26	2.07	2.06	2.18	1.91
Au/Ni/ h -BN		Kagome		Hexagonal	
		$\Delta E_{\text{Hex-Kagome}}$	$z_{\text{BN-Ni}}$	$z_{\text{Au-Ni}}$	$z_{\text{BN-Ni}}$
PBE+vdW	1.35	2.06	2.19	2.21	2.02
LSDA+vdW	1.22	1.94	2.07	1.87	1.85
LSDA	1.35	1.96	2.11	1.93	1.93
Ag/Ni/ h -BN		Kagome		Hexagonal	
		$\Delta E_{\text{Hex-Kagome}}$	$z_{\text{BN-Ni}}$	$z_{\text{Ag-Ni}}$	$z_{\text{BN-Ni}}$
PBE+vdW	1.03	2.03	2.22	2.36	2.09
LSDA+vdW	1.37	1.90	2.08	1.90	1.90
LSDA	1.30	1.94	2.11	1.94	1.96

size of the skyrmions we obtained, which change from a diameter of 5 nm (LSDA) to 22 nm (PBE). Here we find the LSDA results more reliable, given our past experience and the cited literature.

In the light of the previous discussion, we added the following remarks to the description of the Methods in the main text: **The combination of PBE for geometry optimization and LSDA for magnetic properties has been widely used and those works achieve good agreement between theory and experiment [6–11]. Our test calculations show that different functionals do not affect the occurrence of kagomerization and have no significant impact on the magnetic interactions (see Supplementary Note 6).**

As we found that the comment and the reply may be of interest to the readers, we decided to copy our reply including Table R1 and Fig. R1 as a **new note in the Supplementary Material (Note 6: Tests of exchange and correlation functionals)**, with minor adjustments to the wording for readability.

Figure R1: Comparison of magnetic interactions in Pt/Co. a Exchange-stiffness-tensor element A_{xx} . **b** DMI-spiralization-tensor element D_y^x .

Reviewer #1– Comment 3: The statement of rotating the magnetization makes me confused. Does the authors mean rotating only the magnetization directions of every atoms, not others? This will result in wrong ground states. If the plane of two-dimensional kagome lattice is also rotated simultaneously, the distorted skyrmion and the asymmetric bimerons stated by them become incomprehensible.

Authors: We apologize for the confusion we caused. The purpose of this transformation is only to

aid in visualizing the nature of the skyrmionic objects, by comparing to the more familiar point of view of skyrmions in an out-of-plane ferromagnetic background. In the insets of Figs. 5a and b, we applied the same rotation to every spin in order to align the ferromagnetic background along the z -axis. Once this transformation is applied to an asymmetric bimeron, the resulting rotated spin texture is that of a distorted skyrmion, which helps the analysis of our study. In Fig. R2, we illustrate the mathematical rotation of both the symmetric and asymmetric bimeron. In the case of a symmetric bimeron, rotating the spins to align the ferromagnetic background from the y - to the z -axis reveals a standard skyrmion. However, for an asymmetric bimeron, we obtain a distorted skyrmion. The rotated view also helps with determining the spatial extent of these magnetic objects.

The following clarification has been added to the main manuscript: **Note that the uniform rotation applied to all the spins is only used for visualization purposes, in order to aid our subsequent analysis.**

Figure R2: **Mathematical rotation of the spins in the symmetric bimeron and asymmetric bimeron.** **a** From a symmetric bimeron to a skyrmion. **b** From an asymmetric bimeron to a distorted skyrmion.

Reviewer #1– Comment 4: “Au/Co/h-BN” on line 254 should be “Au/Ni/h-BN”.

Authors: We thank the reviewer for pointing out this mistake. It is now corrected.

Reviewer #2: In the manuscript under consideration here, the authors employ first-principles calculations to explore the impact of h-BN capping on transition metal atoms deposited on heavy metals surfaces. Contrary to conventional wisdom, which wants h-BN merely as an inert capping layer, this study unveils the substantial influence that h-BN has on structural, electronic, and magnetic properties.

More specifically, the authors reveal that h-BN serves as a driving force, inducing a structural kagomerization within transition metal monolayers. This process yields electronic dispersions characterized by prototypical kagome features, including Dirac states and flat bands. Furthermore, h-BN induces a significant rearrangement of magnetic properties, evident in modifications to magnetic exchange interactions, the Dzyaloshinskii-Moriya interaction, and magnetic anisotropy energy. These modifications ultimately give rise to transformed magnetic states, leading to the emergence of skyrmions and bimerons, which are absent in the absence of h-BN capping.

The study is conducted with meticulous attention to detail and follows a systematic approach. The main message that h-BN's behavior extends beyond a simple capping role is both intriguing and valuable for the scientific community. However, before recommending publication, it is important for the authors to address the following questions.

Authors: We thank the reviewer for recognizing the significance and importance of our work. In the following we carefully answer all the raised remarks and describe the related changes to the manuscript. We hope that the reviewer will be satisfied with our response.

Reviewer #2– Comment 1: I find it unclear whether the atomic relaxations were carried out in the paramagnetic state or if the authors also considered magnetism. If the calculations were conducted without incorporating magnetism, I expect significant modifications in the hexagonal versus kagome energy difference of Figure 1.

Authors: We thank the reviewer for this comment and apologize for the potential confusion. We have indeed considered the appropriate magnetic states when performing the structural relaxations: the ferromagnetic state for Fe, Co and Ni; the triangular antiferromagnetic Néel state for Cr and Mn; the non-magnetic state for the others. The reported energy differences between the hexagonal and the kagome lattice structures in Fig. 1 take this into account.

This information has now been added to the main manuscript: **We have considered the appropriate magnetic states for the respective 3d elements: the ferromagnetic state for Fe, Co and Ni; the triangular antiferromagnetic Néel state for Cr and Mn; the non-magnetic state for the others.**

Reviewer #2– Comment 2: Figure 2 illustrates a strong hybridization between the kagome-like bands and the heavy metal bands. I am skeptical about the authors' claim that these kagomerized systems provide a new platform for investigating potential correlated topological phases. Furthermore, the kagome lattice is characterized by van Hove singularities exhibiting both pure and mixed sublattice characters, considered pivotal in generating exotic collective orderings observed in kagome metals like AV_3Sb_5 . It remains unclear how the twofold nature of van Hove singularities changes upon deposition on the heavy metal surface and h-BN capping.

Reviewer #2– Comment 3: Are the relatively flat bands observed in Figure 2 panels b) and c) absent in the hexagonal configurations? In other words, are the flat bands a consequence of the kagomerization?

Authors: As **Comment 2** and **3** are related we give a combined answer. We understand the reviewer's concerns and we believe that those concerns can be lifted using our results, as we discuss in the following. To anticipate the conclusion: the kagome physics in the systems that we discuss is distinct from what has been found for the AV_3Sb_5 systems and is driven by flat bands. This difference is already seen in the free-standing transition-metal monolayers and so is not due to the strong hybridization with the surface and/or the h-BN layer. We plan to explore these aspects in the future and share the referee's enthusiasm regarding the potential significance of our findings in this field.

We first recapitulate the basic phenomenology. In the simplest tight-binding model of a kagome lattice we expect to find three types of van Hove singularities (VHS): one from the flat band and two additional ones from constant-energy contours that pass through the M points in the Brillouin zone.

This type of constant-energy contour is also present for a simple tight-binding model of a 2D hexagonal lattice and leads to a single VHS in that case. In a realistic band structure the situation is more complex but we can still use the basic phenomenology for guidance.

In Fig. R3 we compare the band structures of Co monolayers with kagome and hexagonal structures. In the free-standing case with the ideal kagome structure, Fig. R3a, we identify several VHS that can be traced to bands which are very flat across extended portions of the Brillouin zone at -0.8, 0.5, 0.6 and 0.8 eV. These VHS singularities seem to be mostly driven by the very flat band dispersion and it is not clear if the M point mechanism is operative and contributing to them. In contrast, for the free-standing case with the hexagonal structure (Fig. R3d) we find a large VHS that is related to the M point around 0.2 eV, with another VHS close by arising from a fairly flat band around Γ . This sets the stage concerning the VHS without effects from hybridization with the surface or interaction with the *h*-BN layer, and the behaviour is clearly different from the AV_3Sb_5 systems.

We next discuss how the surface and the *h*-BN layer modify this picture, and show that VHS from flat bands do indeed survive and are close enough to the Fermi energy to potentially drive interesting physics. For the kagome case, Fig. R3b and c, we still discern flat bands despite the strong hybridization, such as at -0.15 eV in Pt/Co/*h*-BN (Fig. R3b) and at 0.2 eV in Ag/Co/*h*-BN (Fig. R3c), which lead to VHS close to the Fermi energy (indicated by the arrows in those figures). In contrast, flat bands are not observed near the Fermi energy in hexagonal Pt/Co/*h*-BN (Fig. R3e), while some flat bands survive around Γ for Ag/Co/*h*-BN (Fig. R3f). This is a unique feature of the Ag(111) surface that contains no electronic states in a broad energy region around Γ , preventing hybridization with the Co bands. We conclude that most of the flat bands are not robust against hybridization with the surface electrons, except for those in the kagome lattice near the Fermi energy.

As we found the preceding discussion to be valuable to the reader, we decided to incorporate it in the main text under Results – Dirac cones, flat bands and van Hove singularities.

We have added Kang et al., Nat. Phys. **18**, 301–308 (2022) as a new reference to the manuscript.

Reviewer #2– Comment 4: What is the spatial extent of the skyrmions obtained through atomistic spin dynamic simulations? How does this extension compare to the typical dimensions observed on

Figure R3: Electronic structure of kagome and hexagonal Co monolayers. Spin-resolved Co-projected band structures and DOS for: **a** isolated Co monolayer kagome lattice; **b** kagome structure of Pt/Co/h-BN; **c** kagome structure of Ag/Co/h-BN; **d** isolated Co monolayer hexagonal lattice; **e** hexagonal structure of Pt/Co/h-BN; and **f** hexagonal structure of Ag/Co/h-BN. The arrows indicate some van Hove singularities in the band structures and DOS.

established platforms for skyrmions?

Authors: We thank the reviewer for this valuable question. As shown in Fig. 4 of the main text, the kagomerized structures can host various spin textures. In Au/Fe/h-BN, Pt/Co/h-BN and Au/Ni/h-BN structures, we observed isolated skyrmions and bimerons. In Au/Fe/h-BN, the diameter of isolated skyrmions is about 2.0 nm. In the other two systems, we uniformly rotate the spins by 90° in Pt/Co/h-BN and Au/Ni/h-BN structures to obtain a skyrmion-like view of the spin textures, as illustrated in Fig. R4. In this way we found that the diameter of the bimeron in Pt/Co/h-BN is approximately 13 nm while in Au/Ni/h-BN is about 3.5 nm.

Recent experimental studies have observed isolated skyrmions with diameters down to a few nanometers at cryogenic temperatures. For instance, isolated magnetic skyrmions with a diameter of 5 nm were observed at zero magnetic field in ferromagnetic Rh/Co atomic bilayers on the Ir(111) surface [12]. Under the application of an external field, individual skyrmions with diameters of 2-3 nm emerge in the bilayer of Pd/Fe on the Ir(111) substrate [13]. Consequently, the spatial extent of the

skyrmions and bimerons predicted in our study is of the same order of magnitudes of these small-sized skyrmions experimentally discovered.

The following has been added to the main manuscript: **The spatial dimensions of these topological spin textures in our kagome structures range from about 2 to 13 nm, and skyrmions with similar sizes have been found in related heterostructures [12, 13].**

Figure R4: Determining the diameter of bimerons. **a** Bimeron in Pt/Co/h-BN; **b** Distorted skyrmion obtained from panel **a** through rotating the spins. **c** Bimeron in Au/Ni/h-BN; **d** Distorted skyrmion obtained from panel **c** through rotating the spins.

Reviewer #2– Comment 5: The majority of the results are presented for Ni, Fe, and Co. However, as indicated in Figure 1, the kagome structure is already the most stable for these transition metals even in the free-standing case, rendering kagomerization unnecessary. Notably, no results are provided for transition metals that exclusively favor the kagome lattice through kagomerization. I perceive this as a potential limitation in the study.

Authors: We thank the reviewer for this comment and apologize for the confusion we caused. In the latter part of the manuscript, we concentrate on how the kagomerization influences ferromagnetic properties, focusing on three magnetic elements widely utilized in spintronics: Fe, Co, and Ni. We

chose to do so because the antiferromagnetic case is substantially more complex to analyze. However we hope to do so in a future publication.

Turning to the second part of the comment, in Fig. 1h (HM/TM) and i (TM/*h*-BN) of the main text the kagome (hexagonal) structures originate directly from the relaxed kagome (hexagonal) HM/TM/*h*-BN, removing either the *h*-BN layer or the HM layer without performing any additional relaxation. These structures are synthetic, serving only as an energy comparison to Fig. 1g. In a similar fashion, we present in Fig. 1j the results for free-standing TM monolayers strained to match the lattice constant of three substrates. As the reviewer noticed, the respective total energy differences for elements like Fe, Co, Ni and Cu appear to suggest that the kagome structure is already favored. This in fact highlights the importance of strain in stabilizing the kagome lattice. When we consider the free-standing monolayers and release the epitaxial constraint of matching the lattice constant of the substrate, we find that the total energy differences always favor the hexagonal over the kagome structure at the respective theoretically optimized lattice constants, as shown in Table R2. Besides, for elements like V and Cr, which do not favor kagome lattice even under strain, as shown in Fig. 1j of the main text, the presence of *h*-BN induces the formation of a kagome lattice. This underscores again the pivotal role of *h*-BN as a driving force. Therefore, the kagomerization enabled by *h*-BN which we predict in our manuscript is crucial for realizing the kagome lattice in 3*d* metals.

We added the following clarification to the caption of Fig. 1 in the main manuscript: **The structures in panels h-j are not energy-optimized configurations and serve only for comparison purposes.**

We have also added the following to the main text in connection to Fig. 1: **Note that energy-optimized free-standing monolayers always prefer the hexagonal structure, see Supplementary Note 1.**

We have also included **Table R2 in Supplementary Note 1.**

Reviewer #3

Kagome lattices are emerging as an exciting platform for the rich physics, including magnetism, charge density wave (CDW), topology, and superconductivity. Thus exploring new materials with Kagome lattice structure is a very interesting research.

In this manuscript, based on first-principle calculation, the author search the possible two-dimensional kagome lattice in monolayers of transition metals, such kind of materials is still not exist to my knowl-
 edgement. They find that heterostructures based on h-BN can result in kagomerization.

I think this work is quite interesting and can be accepted by NC.

Authors: We are grateful to the reviewer for recommending the publication of this manuscript in *Nature Communications*. We believe that our prediction of kagomerization assisted by *h*-BN offers new opportunities for further experimental exploration on highly-sought kagome lattices and the underlying intertwining of topology, electron-correlations and magnetism.

Authors: Finally, we would like to thank all the reviewers again for their valuable suggestions and comments. We hope that the revised manuscript is now appropriate for publication in *Nature Communications*.

Table R2: Energy differences $\Delta E_{\text{Hex-Kagome}}$ (eV) between the freestanding hexagonal and kagome structures of the TM monolayer with relaxed lattice constants (\AA).

	Sc	Ti	V	Cr	Mn	Fe	Co	Ni	Cu	Zn
$\Delta E_{\text{Hex-Kagome}}$	-1.51	-2.33	-2.38	-1.72	-1.42	-2.09	-1.97	-1.93	-1.41	-1.04
a_{Hex}	5.43	4.63	4.23	4.58	4.50	4.16	4.05	4.08	4.19	4.41
a_{Kagome}	6.00	5.04	4.56	5.14	4.96	4.74	4.51	4.57	4.71	5.00

References

- [1] Ajejas, F. *et al.* Thermally activated processes for ferromagnet intercalation in graphene-heavy metal interfaces. *ACS Appl. Mater. Interfaces* **12**, 4088–4096 (2020). URL <https://doi.org/10.1021/acsami.9b19159>. PMID: 31875389, <https://doi.org/10.1021/acsami.9b19159>.
- [2] Decker, R. *et al.* Local tunnel magnetoresistance of an iron intercalated graphene-based heterostructure. *J. Phys.: Condens. Matter* **26**, 394004 (2014). URL <https://dx.doi.org/10.1088/0953-8984/26/39/394004>.
- [3] Decker, R. *et al.* Atomic-scale magnetism of cobalt-intercalated graphene. *Phys. Rev. B* **87**, 041403 (2013). URL <https://link.aps.org/doi/10.1103/PhysRevB.87.041403>.
- [4] Cano, B. M. *et al.* Rashba-like spin textures in graphene promoted by ferromagnet-mediated electronic-hybridization with heavy metal (2023). 2206.04351.
- [5] El-Kerdi, B. *et al.* Evidence of strong Dzyaloshinskii–Moriya interaction at the cobalt/hexagonal boron nitride interface. *Nano Lett.* **23**, 3202–3208 (2023). URL <https://doi.org/10.1021/acs.nanolett.2c04985>. PMID: 37053437, <https://doi.org/10.1021/acs.nanolett.2c04985>.
- [6] Romming, N. *et al.* Competition of Dzyaloshinskii-Moriya and higher-order exchange interactions in Rh/Fe atomic bilayers on Ir(111). *Phys. Rev. Lett.* **120**, 207201 (2018). URL <https://link.aps.org/doi/10.1103/PhysRevLett.120.207201>.
- [7] Spethmann, J. *et al.* Discovery of magnetic single- and triple- \mathbf{q} states in Mn/Re(0001). *Phys. Rev. Lett.* **124**, 227203 (2020). URL <https://link.aps.org/doi/10.1103/PhysRevLett.124.227203>.
- [8] Li, W., Paul, S., von Bergmann, K., Heinze, S. & Wiesendanger, R. Stacking-dependent spin interactions in Pd/Fe bilayers on Re(0001). *Phys. Rev. Lett.* **125**, 227205 (2020). URL <https://link.aps.org/doi/10.1103/PhysRevLett.125.227205>.
- [9] Ferriani, P. *et al.* Atomic-scale spin spiral with a unique rotational sense: Mn monolayer on W(001). *Phys. Rev. Lett.* **101**, 027201 (2008). URL <https://link.aps.org/doi/10.1103/PhysRevLett.101.027201>.

- [10] Heinze, S. *et al.* Spontaneous atomic-scale magnetic skyrmion lattice in two dimensions. *Nat. Phys.* **7**, 713–718 (2011). URL <https://doi.org/10.1038/nphys2045>.
- [11] Nickel, F. *et al.* Coupling of the triple- \mathbf{q} state to the atomic lattice by anisotropic symmetric exchange. *Phys. Rev. B* **108**, L180411 (2023). URL <https://link.aps.org/doi/10.1103/PhysRevB.108.L180411>.
- [12] Meyer, S. *et al.* Isolated zero field sub-10 nm skyrmions in ultrathin Co films. *Nat. Commun.* **10**, 3823 (2019). URL <https://doi.org/10.1038/s41467-019-11831-4>.
- [13] Romming, N., Kubetzka, A., Hanneken, C., von Bergmann, K. & Wiesendanger, R. Field-dependent size and shape of single magnetic skyrmions. *Phys. Rev. Lett.* **114**, 177203 (2015). URL <https://link.aps.org/doi/10.1103/PhysRevLett.114.177203>.

REVIEWERS' COMMENTS:

Reviewer #1 (Remarks to the Author):

The replies of the authors clarify my problems, so I recommend the publication of this manuscript in Nature Communications.

Reviewer #2 (Remarks to the Author):

The authors have carefully addressed my concerns from the first round. The paper can now be accepted for publication. My only remaining concern is that a divergence in the density of states from a flat band cannot be, technically speaking, called a van Hove singularity.

Reviewer #3 (Remarks to the Author):

Quickly reading the comments from other Referee, and the response from the authors. I think the authors properly answer the questions, thus I think this manuscript can be accepted by NC.